# Exploring Open Dialogue and autism: A qualitative client-perspective study

Karin Lorenz-Artz [1,2]*, Joyce Bierbooms[1], Marieke Kneepkens[2], Inge Bongers[1]

1 Tranzo, Tilburg School of Social and Behavioral Sciences, Tilburg University, Tilburg, The Netherlands,
2 Mental Health Care Institute Eindhoven, Eindhoven, The Netherlands

* c.a.g.lorenz@tilburguniversity.edu

## Abstract

Mental health care for people diagnosed with autism is transforming towards recovery-oriented, person-centered network care. Within this changing landscape, the Open Dialogue approach is acknowledged as a promising approach that exemplifies this transformation. Characterized by its transparent and equitable collaboration between client, their network and mental health care professionals, the Open Dialogue approach shows promise for people diagnosed with autism but also poses potential challenges, leaving uncertainty regarding its suitability. Given these potential benefits and challenges, the present study aims to explore how clients diagnosed with autism experience the Open Dialogue approach firsthand. This qualitative study explores how 12 participants diagnosed with autism experienced the OD approach at a specialized mental health care center for autism. The study employs six in-depth interviews and two focus groups, utilizing a hybrid analysis approach that combines both inductive and deductive thematic techniques. Findings reveal that participants generally have positive experiences with the Open Dialogue approach, highlighting key elements such as connectedness, reassurance, recognition, agency, and support through collaborative interaction. However, a minority reported negative experiences linked to feelings of uncertainty and discomfort with therapist reflections. These findings suggest that people diagnosed with autism evaluate the Open Dialogue approach in a similar way to other mental health service users. Despite the potential challenges posed by Open Dialogue, our results indicate that providing Open Dialogue care may be valuable for people diagnosed with autism, as it is for other mental health service users. Further research is needed to substantiate this indication.

## 1. Introduction

The mental health care (MHC) landscape for people diagnosed with autism is undergoing a transformative shift, with the recovery approach, alongside person-centered and network-centered strategies, being recognized worldwide as one of the key

**Data availability statement:** The datasets presented in this article are not readily available because the raw data supporting the conclusions of this article cannot be made completely anonymous. Requests to access the datasets should be directed to Karin Lorenz, c.a.g.lorenz@tilburguniversity.edu and/or institutional review board of MHC Eindhoven and Kempen (GGzE), wetenschapscommissie@ggze.nl.

**Funding:** The authors received no specific funding for this work.

**Competing interests:** The authors have declared that no competing interests exist.

**Abbreviations:** MHC, Mental health care; OD, Open Dialogue.

pillars [1–3]. Equal collaboration between clients, their social network, and care professionals is considered essential for promoting hope and empowering people. This shift moves the focus from diagnosing and intervening to adopting a more holistic approach [4,5]. Several authors suggest that this collaborative approach enhances the quality of care and better meets individual needs [6,7]. Furthermore, it empowers clients through shared decision-making and active participation in their treatment [8–10]. In this context, clinicians act more as guides rather than as experts [11–13].

Within this evolving landscape, the Open Dialogue (OD) approach has emerged as an acknowledged and promising approach that effectively applies recovery-oriented, person-centered, and network-based care [14–16]. OD promotes transparent, open, and equal collaboration among clients, their social network and professional care network in so called network meetings. OD's primary focus is on ensuring that each involved member feels heard and respected, shifting from a solution-oriented perspective to a relation-oriented perspective [17,18]. Central to OD is the shared emotional experience between people in a "dialogic practice", a distinctive therapeutic interaction based on twelve fundamental elements [18] (see Table 1). This approach emphasizes connecting, shared meaning-making and exploring experiences rather than diagnosis or treatment plans. Connecting and reconnecting people is not an a-specific factor within OD, as it is often considered. Instead, it can be regarded as the specific driving force of this approach [19]. The dialogical process can exert a therapeutic effect right from the start, enabling each member of the network to experience a sense of personal autonomy, as well as interdependence [17]. This (inter)person-centered foundation and emphasis on the client's capacity and environment to take control of their personal recovery align with the principles of the recovery vision [20].

Since the 1990s, favorable outcomes associated with the OD approach have been reported in Western Lapland [21]. For example, this study showed that 82% of individuals experiencing acute psychosis were symptom-free five years after receiving OD treatment. Additionally, 86% had resumed full-time work or studies, with only 14% relying on disability allowance. A subsequent study revealed that over 80% of clients treated with the OD approach were fully employed or enrolled in education within two years [22]. A long-term study spanning nineteen years confirmed that many of the favorable outcomes observed in earlier research remained stable over time [23,24].

In recent years, OD has received considerable attention and is now being implemented in multiple countries for various client groups while undergoing scientific research [e.g., 25–27]. Despite the growing interest, it has also been noted that there is still a lack of high-quality evidence to fully substantiate the long-term benefits of Open Dialogue [26]. However, several studies have shown that clients perceive the OD approach as distinctive from conventional treatments, highlighting its emphasis on transparency and so called reflection moments. During these reflection moments, participants listen attentively to the therapist's reflections without an immediate response. Overall, they appreciate the OD approach for fostering constructive relationships characterized by trust, respect, accessibility, and a collaborative foundation, in addition to its

**Table 1. The 12 basic elements of OD [18].**

| Elements | Short description |
|---|---|
| *Two (or more) OD therapists are present* | This gives shape to the reflective process within OD, a moment in which the OD therapists collectively reflect on what they are experiencing in the session, with the client and the network being witnesses. |
| *Participation of social network* | The client (or requester of the network meeting) is asked who is important to involve in the meeting and if, where possible, the client could do this themselves. This way, equality is introduced right away. |
| *Using Open-Ended Questions* | Both the standard questions at the beginning of the session ("What is the background of this meeting?" and "How do you want to use this meeting?") and the questions during the meeting facilitate each participant in expressing their own thoughts and feelings, allowing them to highlight what is specifically important to them. |
| *Responding To Clients' Utterances* | By repeating words from attendees, allowing for silence, and not directing the conversation in a particular direction, space is created to bring forth everything that the client and the network want to communicate, including what has not yet been expressed (in words or non-verbally). This attentive listening stance requires the therapists to be constantly "present." |
| *Emphasizing the Present Moment* | Reflecting on the here and now and on emerging (intense) emotions, without immediately assigning them meaning, provides a safe way to bear and allow emotions to exist. |
| *Eliciting Multiple Viewpoints* | The polyphony (multiplicity of voices) that plays a significant role within OD ensures equality among participants: everyone's story is equally important. The therapists do not aim for everyone to agree but encourage sharing and exploring different viewpoints, experiences and voices, even when they might conflict. |
| *A relational focus* | OD operates within a relational framework, where all subjects have a relational aspect. By focusing on this, new meanings can emerge. |
| *Considering the presented issues as meaningful* | Labeling problems or symptoms as 'logical' and 'understandable' within the context in which they occur has a normalizing character. It reduces the risk of polarization between 'ill' and 'healthy'. |
| *Emphasizing clients' stories, not symptoms* | By paying attention to one's own story and reflecting on significant personal details, and by expressing these in words, a shared understanding can emerge. This creates new opportunities for connection. |
| *Therapists' reflections in the meetings* | While the client and the network listen, the therapists discuss among themselves what they are experiencing and what comes to mind. Only after they have spoken, are the client and their network invited to respond. |
| *Being Transparent* | Everything is discussed in the presence of the client. It makes the client and their network full partners in all important decisions. |
| *Tolerating Uncertainty* | The therapists do not operate with a preconceived plan or goal, and tolerating uncertainty prevents hasty decisions. The therapist does not position themselves as an expert and provides both time and space for all involved parties to arrive at a shared sense-making. |

commitment to transparency [e.g., 28–30]. However, the question remains whether this promising recovery-oriented OD approach is suitable for all client groups [25,26].

Now, by threading together the question whether OD is suitable for all client groups and ODs core of nurturing connectedness, we direct our gaze specifically on people diagnosed with autism. Since for these individuals fostering connections is often at the core of their challenges, it's precisely in this aspect that OD can hold great potential. As we venture deeper into these challenges, adults diagnosed with autism often face challenges such as limited social integration, poor employment prospects, and a high prevalence of mental health problems [31]. They have an increased risk of developing psychological issues [32,33], including suicidal feelings and behaviors [34–36]. Feelings of loneliness and reduced perceived

social support have been associated with suicidality, depression, and lower quality of life [37–40]. OD seamlessly aligns with these challenges because at its core, OD is about connectedness, equality, and personal empowerment.

On the other hand, besides ODs great potential, there are also indications that suggest OD may not align well with individuals diagnosed with autism, such as the required skill of 'tolerating uncertainty,' which applies not only to the OD therapists but also to the client and their social network [41]. Individuals diagnosed with autism often report that even the slightest uncertainties and ambiguities can unsettle them and lead to feelings of anxiety [42]. Another indication that the OD approach may pose difficulties for individuals diagnosed with autism is that autism is characterized, among other things, by specific social communication difficulties [43]. These difficulties include initiating conversations, taking turns, dealing with abstract language, and interpreting non-verbal cues [44], which are integral to a dialogical process [18]. Additionally, the size of the network meeting can also be challenging due to communication difficulties and sensory sensitivity [43]. These concerns are reflected in some previous studies which have shown that certain clients perceive the OD approach as burdensome due to difficulties in accepting uncertainty, such as the absence of a preset agenda prior to the meeting, and the lack of specific advice or solutions [29,30,45]. Nonetheless, the literature does not provide a clear conclusion on the suitability of OD to individuals diagnosed with autism.

Even though the literature does not provide a clear answer, professionals in a specialized mental health practice for autism in the Netherlands were inspired by promising results from Finland [21–24]. Seeing potential in the OD approach for individuals diagnosed with autism, they began offering OD care specifically for this group. They integrated OD network meetings into their regular care, which provided the opportunity to gain more insight through practice-based research into the promise and potential challenges of OD for individuals diagnosed with autism, and its alignment with the needs of clients diagnosed with autism. Therefore, the aim of the current study is to explore how individuals diagnosed with autism experience the OD approach themselves. Gaining a deeper understanding of their experiences can illuminate important themes related to recovery-oriented care for individuals diagnosed with autism and gives insight into the suitability of the recovery-oriented OD approach for people diagnosed with autism.

## 2. Method

### 2.1. Ethics statement

This study was approved by the institutional review board of MHC Eindhoven and Kempen (GGzE), which conducted an ethical and feasibility assessment (reference ILB/2020051).

### 2.2. Setting

This study was conducted at the ambulatory Autism Center of MHC Eindhoven and Kempen (GGzE), a specialized MHC organization in the southern part of the Netherlands. This organization provides support to individuals with severe and often persistent psychiatric and psychosocial problems.

All clients treated at the ambulatory Autism Center are adults diagnosed with autism, with intelligence levels ranging from average to high. They are generally capable of verbal expression, although their verbal communication skills may vary. The multi-disciplinary treatment team (including, e.g., clinical psychologist, case managers, nurse practitioner and mental health social workers) actively tailors its communication to each client, using a flexible and sometimes playful approach that adapts to the client's communicative abilities.

Treatment as usual involves regular meetings with a coordinating clinician and a case manager to develop an individualized treatment plan. Outpatient care may include support with daily structure, daytime activities, financial matters, and psychological treatment for comorbid symptoms such as anxiety. Involvement of a client's social network is possible but not required.

Clients can initiate OD network meetings themselves, but in practice, such meetings are usually initiated by the treatment team. In principle, all clients under the care of this outpatient team are eligible for OD. When the team or an

individual professional observed or anticipated a crisis (e.g., when a client was off balance or an impasse had emerged in treatment), or noticed unrest within the client's network, the client was offered the opportunity to participate in an OD network meeting. OD was considered helpful in restoring communication, rebuilding trust, and creating new possibilities for moving forward.

Within the treatment team, six professionals were trained to provide OD care. As a result, it was not feasible to offer OD to all clients, and it was prioritized for situations with the greatest need. These six OD therapists successfully completed the one-year postgraduate training program called "Peer-supported Open Dialogue, Social Network and Relationship Skills" at the Academy of Peer-supported Open Dialogue (APOD) in the United Kingdom, which is accredited by the London South Bank University.

When OD network meetings were integrated into regular treatment, the OD therapists worked in fixed pairs throughout the client's treatment trajectory, following the twelve basic elements of "dialogic practice" as outlined in Table 1 [18]. Clients had full autonomy to decide whether they wished to participate and whom they wanted to invite to these meetings. The OD network meetings were held in person, with no predefined agenda. At the beginning of each meeting, attendees were always asked how they wanted to use the meeting time. Examples of topics discussed included challenges in collaboration with professionals, issues arising in relationships between family members, and difficulties in deciding on further treatment options.

## 2.3. Design

In this exploratory study, we employed a qualitative multi-method design that combined semi-structured individual interviews and focus groups [46]. The two methods were used for methodological triangulation: focus groups elicited interactional sense-making and participants' experiences with the OD approach, while interviews enabled participants to discuss sensitive personal experiences in a confidential setting [47]. To maintain independence between datasets and prevent carry-over, participants were allocated to one modality only (focus group or interview). The initial plan was to conduct focus groups first, followed by individual interviews to explore specific aspects in more depth. In practice, recruitment constraints required early applicants to be invited for interviews first, while later applicants participated in focus groups. Despite this change in sequence, the combination of interviews and focus groups ensured comprehensive coverage of the research questions, and no inconsistencies were observed in the data.

Data collection and analysis proceeded iteratively, using a hybrid thematic analysis that combined deductive and inductive approaches [48]. This strategy leverages the strengths of both techniques to provide a more comprehensive grasp of the data [49]. Deductive coding was guided by key concepts from the OD literature, while inductive coding allowed themes to emerge from participants' accounts of their experiences with OD in the context of an autism diagnosis. Given the existing literature on OD, but the limited knowledge of how clients diagnosed with autism experience this approach, it was important to apply both analysis approaches.

Saturation was assessed pragmatically by monitoring whether additional interviews or focus groups generated new codes or themes relevant to the research questions. Toward the end of the study, no substantial new insights were identified, indicating that saturation had been achieved. At the same time, data collection was constrained by the limited availability of one of the researchers. Had saturation not been observed, the OD-trained researcher would have re-approached participants who had been unable to attend the second planned focus group and invited them to take part in individual interviews. Thus, while practical considerations also contributed to ending recruitment, our conclusion that saturation had been reached was ultimately based on the analytic process.

Each focus group was moderated by a member of the research team with a background in autism and extensive experience working with individuals diagnosed with autism and conducting sensitive, in-depth conversations. Importantly, this moderator was not involved in the care of any participant. She was supported by a second researcher who was trained in the OD approach and experienced in qualitative interviewing. This second researcher managed logistics,

took observational notes, and, where appropriate, contributed to the discussion to deepen the exploration of participants' perspectives. The moderator followed a semi-structured topic guide, encouraged balanced participation, and emphasized confidentiality and psychological safety throughout.

## 2.4. Recruitment and data collection procedure

### 2.4.1. Participants.

All clients who attended at least one OD network meeting during the period from December 1st, 2020 and April 30th, 2022 were eligible for this study. This group of eligible clients experienced severe and often persistent psychiatric and psychosocial problems, having been diagnosed with autism and possessing an intelligence level ranging from average to high. While they were capable of expressing themselves verbally, there were variations in their communication skills. Additionally, a (risk of) crisis was present at the time the treatment team offered the client the opportunity for an OD network meeting. This involved either a loss of balance for the client or an impasse in treatment, or unrest within the client's network.

### 2.4.2. Recruitment.

After informing all OD therapists at the Autism Center about the study during a regular monthly meeting, three recruitment strategies were implemented. First, after each OD network meeting, OD therapists provided clients with an information letter and asked if the researcher could initiate contact. To ensure maximum variation, we included clients who had expressed a desire not to participate in a second network meeting (drop-outs). Second, the secretary contacted clients who had participated in an OD network meeting within the past six months to ask if they were open to being approach for the study. Those who agreed were sent an information letter, and the researcher contacted them by phone. Third, flyers and information letters were placed in the waiting room, and interested clients were given one week to decide. After this period, the researcher scheduled a meeting to obtain informed consent, explaining the study procedure. Written consent was obtained from all participants prior to scheduling the date for the interview and focus group.

Between December 1st, 2020 and April 30th, 2022, 36 clients were approached, of whom 12 (five females and seven males) participated in the study. Their ages ranged from 25 to 50 years, with an average age of 40 years. Four out of the twelve participants were employed, seven were unemployed or had no daily occupation, while one participant was a student. All of them had prior experience with other (therapeutic) sessions within the MHC. On average, participants had two OD network sessions, ranging from one to five sessions, which is representative of the client group at the studied site. In addition to the OD therapists and the participant themselves, an average of five network members were present during the OD network sessions.

Although we initially intended to start with a focus group, recruitment proved challenging due to a limited number of applications. Concerned about participants losing interest while waiting for a full group to form, we decided to invite them for individual interviews instead. Six individual interviews were conducted, lasting approximately one hour each. Afterward, we received a cluster of applications in quick succession, which presented the opportunity to create two focus groups of six participants per group. We approached these individuals to gauge their interest in participating in a focus group, and all expressed their willingness to do so. If they had not wanted to participate in a focus group, we would have provided them with the option to participate in an individual interview. In the first focus group, two participants were absent, one with an unknown reason and the other due to last-minute COVID-19 concerns. In the second focus group, four out of six participants were absent. Due to logistical and time constraints associated with study requirements of one of the involved researchers, we could only approach the absent participants from the first group after its completion. Importantly, participants were assigned to only one data collection modality: those who did not participate in a focus group could later be invited to an individual interview, but no participant took part in both a focus group and an interview. This ensured that the datasets remained independent and that interview responses were not influenced by prior participation in focus groups. Unfortunately, we were unable to involve those who missed the second focus group. As a result, the first group had four participants, and the second had two, with each session lasting about two hours.

**2.4.3. Semi-structured interviews and focus groups.** To gain insight into the participants' experiences with the OD approach, we employed a predefined topic list. These topics were determined through discussions with OD therapists at the Autism Center and existing literature on OD [18, Table 1]. These topics were compiled into a list and used by the researchers as a memory aid for both the interviews and focus groups, to ensure that all relevant aspects necessary to address the research question were included. Both individual interviews and focus groups started with open-ended questions about participants' experiences with the OD approach, such as "How did you come across this new way of working?", "Can you tell us what you think the added value of this approach is?", and "Can you tell us about anything you found less pleasant or experienced as a challenge?". The researchers then responded to the participants' answers to encourage them to share their views and speak freely about aspects they considered relevant. Finally, any topics from the list that had not yet been discussed were addressed. For example, if the participant did not mention OD's adage "nothing about us, without us", the researcher would bring up the topic by saying, "In OD, the principle is: 'Nothing about the client, without the client.' In other words, everything discussed happens in your presence. Can you share how you experience that?" Both individual interviews and focus groups were conducted in person.

The research team consisted of four members: two researchers and two supervisors. While all team members had prior knowledge of autism and OD, each contributed their own expertise to the research. One researcher specialized in autism, while the other focused on OD. The supervisors brought extensive experience in qualitative research within the field of mental health care. Interviews were conducted by the researcher with the background in autism, who has extensive experience working with individuals diagnosed with autism and conducting sensitive, in-depth conversations. It is important to note that participants were not under her care. We believed that her expertise helped us gain profound insights from the participants. We chose a single interviewer to foster a confidential and trusting environment. For the focus groups, both researchers were involved to enhance the recognition of non-verbal cues, ensure balanced participation, and improve data quality reliability through cross-verification of observations and notes.

## 2.5. Data analyses

The interviews and focus groups were audio-recorded, transcribed verbatim and analyzed using the qualitative data analysis program called Atlas.ti (www.atlasti.com, accessed on 14-3-2024). We adopted a hybrid iterative approach, conducting an analysis that incorporated both a deductive perspective (theory-driven, rooted in the OD perspective) and an inductive perspective (data-driven, specifically centered on the experiences of individuals diagnosed with autism). Our analysis adhered to Braun and Clarke's six-phase thematic analysis framework [46]. These six phases consist of becoming acquainted with the data, generating initial codes, identifying themes, reviewing and defining themes, and ultimately capturing and describing these findings in the final report [48]. In the initial coding phase, we utilized codes derived from the topic list containing OD concepts (deductive), including reflection moment, nothing about me without me, network involvement, polyphony. Additionally, codes were generated through an inductive approach based on the new information, particularly from the perspective of participants diagnosed with autism. This included codes like the need to go straight to the point, seating arrangements, and the need for clarity. In this initial coding phase, the interviews and focus groups were coded separately. The researchers then compared the codes from both groups and concluded that they were similar, indicating that the data from the focus groups did not fundamentally differ from the interviews. As a result, it was decided not to distinguish between the data from the two after this initial coding. However, it was always possible to trace back each code to its source, whether it was from an interview or a focus group. In the subsequent analysis phases, the researchers did not encounter any significant issues that would require additional bias checks. To make sense of the resulting nearly 400 codes, we chose to cluster them on a timeline as an interim step: the start of the network meeting, during the network meeting, and the end of the network meeting. Subsequently, we organized codes by clustering or dividing them, renamed fragments, and main and sub-themes were formulated with the OD theoretical background as our guiding compass, given our central question regarding how individuals diagnosed with autism specifically experience this approach. This led to

the emergence of the central themes, which we described in the results section. We concluded by evaluating the degree to which these themes addressed our research questions and proceeded to describe the findings, which are detailed in the results section. During the analysis, the researcher specialized in autism took the lead, as we considered it important that the responses of clients with autism were interpreted appropriately, with an emphasis on inductive analysis. The researcher with expertise in OD was closely involved throughout the entire coding and writing process to ensure accurate interpretation of the OD approach, with a more deductive focus. Code generation and theme development were discussed extensively to reach consensus. The supervisors were involved at predetermined intervals and were also consulted by the researchers whenever uncertainties arose.

## 3. Results

In this results section, we describe the themes that illustrate how individuals diagnosed with autism experienced the OD approach. Analysis of the data reveals that ten out of the twelve participants had a positive perception of the OD approach. Two participants expressed pronounced negativity towards various aspects of OD.

The central themes in participants' experiences were (1) the sense of connectedness, (2) the sense of reassurance, recognition and agency, (3) the sense of support in collaborative interaction, and (4) the sense of uncertainty.

### 3.1. Sense of connectedness

The majority of participants experienced togetherness and equality related to a sense of connectedness, which is central to the OD approach. Participants described the importance of this togetherness with regard to their collaboration with family members and professionals. Furthermore, it was mentioned that it is pleasant to collectively shape a narrative and determine a direction or purpose. In this light, one of them expressed that it alleviated tension because he felt a shared responsibility to collectively make the right meaning of his story. Another participant emphasized that it instills trust when the core is touched upon collectively. Feeling understood creates more space to receive proposals from others and perceive them as less threatening.

*"It has to do with trust because everyone is present, and it is described in the right way. You think, 'Yes, that's exactly it, how I feel, and everyone is on the same page.' And that builds trust, and because you have that trust, knowing that your environment is aligned with you, when they suggest something like 'shall we do it this way or that way?' It's no longer as threatening." (participant 8)*

In addition, several participants mentioned experiencing a pleasant sense of equality as human beings in their interactions with the OD therapists, emphasizing the human element in this therapeutic relationship. They felt that the OD therapist did not present themselves as a therapist but as a fellow human being. They did not feel labeled as clients or 'problems,' and the focus was not on the goals to be achieved but on the interpersonal connection between people. They noted that it provided space to discuss their experiences regarding their collaboration, such as the difficulties they experienced as clients in their interactions with their therapist (e.g., the therapist consistently arriving late).

*"It helps me tremendously to see the person behind the job, because.... I connect with people, not targets, so to speak. In the sense of…sure I want to work on myself, but I gain much more from having a genuine person sitting here in front of me, rather than someone who wants the best for me within that role…" (participant 9)*

### 3.2. Sense of reassurance, recognition and agency

The majority of participants described a feeling of reassurance, recognition, and agency, with the transparency of the OD therapists playing a crucial role. This transparency was particularly evident during reflection moments and was

predominantly positively appreciated for the sense of recognition it provided. Participants also mentioned that it gave them a sense of reassurance and agency, as everything was discussed in their presence, and every person present heard the same accurate reflection. These reflection moments were a surprise to all participants, as they were not familiar with such practices in other counseling conversations.

*"It's like you just continuously get a certain reassurance in the conversation of 'okay, you know, nothing is happening behind the scenes or among themselves. Everything is laid out in front of you'." (participant 9)*

Additionally, several participants noticed that the network meeting took place in a seating arrangement or circle without a table in the middle. This seating arrangement was considered refreshing, contributed to transparency, and created an informal atmosphere. Although some initially found this arrangement uncomfortable, the discomfort quickly disappeared due to habituation and the presence of familiar individuals. However, for some it gave a sense of exposure, which was found uncomfortable because it gave the impression that they couldn't hide behind anything.

*"Yes, it just feels more open without tables, so I think there might be something to that, that it can be kept that way." (participant 8)*

*"To sit in a circle all together and be the subject of conversation myself, I find it terrifying." (participant 2)*

In addition, not all participants experienced this feeling of reassurance, recognition, and agency. For these participants, the reflection moments had limited or a rather counterproductive effect and were therefore not reassuring. For example, one participant didn't understand why the OD therapists were so emotionally affected, as the problem wasn't theirs, and found the reflections to be vague. Another participant explained that she did not feel acknowledged and understood. She expressed feeling rather embarrassed by the therapists' expressions during the reflection moments. This remained a hindrance throughout the remainder of the network meeting. Another participant also mentioned that the emotion or opinion of the OD therapists triggered a fear of having done something wrong and the need to reassure the caregiver. This, in turn, resulted in a tendency to reverse roles with the OD therapists.

*"I found it really strange. I think, 'I should indeed listen to what they think about this conversation,' and I am very sensitive to the opinion of others. Indeed, the only thing that goes through my head is 'am I doing it right? Am I doing it wrong? Did I say something wrong?' And then I hear that they get an anxious feeling, so they start expressing their feelings. I can't do anything with that. […] I used to be a caregiver, so then I start thinking, 'how can I help you again?' So I reverse the roles. […] I'm old school; you're my caregiver after all, and indeed, I don't need to hear your emotions. I find that strange. You can indeed summarize, clarify, ask good questions, summarize again, explain something. That's my view." (participant 2)*

### 3.3. Sense of support in collaborative interaction

Participants experienced an enhanced relational interaction within the network meeting, related to a sense of support in collaborative interaction from the OD therapists. Participants highlighted the potential added value of the therapists' responsive attitude and relational focus, particularly for individuals diagnosed with autism. The therapists' responsiveness extended to picking up on non-verbal cues from all participants and encouraging them to express themselves. The relational focus facilitated the sharing of emotions and experiences, often through questions like, "And how does that feel for you?". This approach was found to be supportive for individuals diagnosed with autism. Moreover, several participants valued the independent stance of OD therapists posing open-ended questions that provided a fresh perspective. This approach fostered the emergence of novel ideas beyond established frameworks.

*"Sort of like a facilitator who notices when someone is listening and their eyes wander, like 'yes' or something, and then 'you have a different opinion' or something. That... yeah, I think that's very good, especially for people with, for example, autism or people who have difficulty expressing themselves, that there's someone there who looks at facial expressions or body language. Someone who ensures that people who have tried to jump into the conversation five times already, eventually say something and things like that." (participant 3)*

Participants noted that the reflection moments provided "here-and-now feedback," offering valuable guidance for shaping the ongoing interactive process. By introducing a pause, the spotlight is placed on what is happening in the very moment during the network meeting, and this is communicated to everyone involved. When the reflection touched upon the essence, it is perceived as constructive. Although it can be challenging at times, it serves as a starting point for progress according to participants.

*"They then discuss 'do you see what's happening here?'. Then they get a kind of real-time feedback, like: what conversation patterns are emerging?" (participant 9)*

Furthermore, participants mentioned two other related aspects that can influence the sense of support in collaborative interaction and are often challenging for people diagnosed with autism: social communication and sensory processing. One participant noted that focusing on the conversation helped him to cope with sensory stimuli from the group, reducing distractions. Additionally, two participants found having a loved one with them to be supportive in managing these challenges. However, for two participants, social communication and social stimuli posed substantial difficulties.

*"In hindsight, of course, it was always a lot. It had been busy, a lot of information, who said what, and I don't know what else. That's why I found it very pleasant that I wasn't alone." (participant 3)*

### 3.4. Sense of uncertainty

Most participants indicated experiencing uncertainty regarding the purpose, content, and conclusion of the conversation. Remarkably, the majority of the participants effectively managed this uncertainty. Nevertheless, for two participants, this sense of uncertainty appeared to have played an important role in generating negative feelings toward the OD approach. Additionally, the sense of uncertainty seemed to play also a role in how participants perceived the group size.

Regarding the lack of clarity before the network meetings, participants mentioned experiencing uncertainty. They didn't know what to expect and how an OD network meeting unfolds (for example, the use of characteristic reflections). Two participants were distinctly negative. They found the lack of clarity regarding the purpose and content of the conversation unpleasant. One participant recommended making the substantive preparation for clients more concrete, which may also increase agency: clients can think about what they want to discuss in the network meeting, preventing them from feeling overwhelmed.

*"Prepare the client, like 'Well, we're going to do this.' Help the client... 'Is it better for you to write down in advance what you want to talk about? Think about it in advance.' Then you give a bit of agency as homework... But in a network meeting, it's too overwhelming. […] No open end, please, but no open beginning either. I just need a red thread. It has to be predictable for me in advance. […]" (participant 2)*

In a similar vein, several participants mentioned that they appreciated when a network meeting concluded with a clear outcome. They noted that it was nice to leave with concrete agreements and goals or a sense of direction to follow, which

helped them regain a sense of control. The absence of a clear conclusion, coupled with a feeling of sensory overload, left another participant feeling uneasy when leaving the session.

> *"[...] because I just felt that everything was aligned in that network meeting, so we all headed in the same direction. That made me feel a lot better." (participant 4)*

> *"But indeed, because I hadn't closed it in my mind yet. So I was still there with a full head and loose ends. So I think: 'Yeah, therapists, help me now.' They were already gone..." (participant 2)*

Now, shifting our attention to the group size, it became evident that the participants' perception of the group size was influenced by the clarity of the purpose for their attendance. To illustrate, one participant had a positive experience when they invited all the professionals involved, ensuring that everyone heard the same story simultaneously. On the other hand, the absence of a clear objective during the network meeting resulted in a negative perception of the group size for another participant.

> *"That also feels a bit stronger when you can say that to a whole team. To all the clinicians at once: 'I'm struggling with this,' compared to having to explain yourself to someone different every time... Having to explain 'why does it bother you?' and 'what are you struggling with?' to someone new each time." (participant 3)*

## 4. Discussion

The paradigm of MHC for individuals diagnosed with autism is evolving towards a recovery-oriented model that emphasizes person-centered and network-based approaches. In light of the overarching question regarding the suitability of the promising recovery-oriented OD approach for this group, this study explored the firsthand experiences of individuals diagnosed with autism who participated in OD network meetings. A deeper understanding of these experiences can shed light on the suitability of a recovery-oriented approach such as OD for this population.

Our results showed that individuals diagnosed with autism generally reported positive experiences with the OD approach, aligning with findings from other mental health service users in previous studies [29,30,45]. They described feelings of connectedness, reassurance, recognition, agency, and support in collaborative interaction, which are elements widely acknowledged in both recovery and therapeutic alliance literature [e.g., 50–55]. However, a minority reported discomfort, similar to previous research [29,30,45]. These individuals reported finding the reflections uncomfortable and awkward, and they also expressed concerns about the uncertainty associated with unclear goals and outcomes of the OD network meetings.

A prominent theme that emerged from the findings was a sense of connectedness (a feeling of belonging) and togetherness (not feeling alone in one's struggles). Participants felt more deeply connected to both their personal network and the OD therapists. The sense of connectedness and togetherness is universally recognized as a fundamental human need. Its absence has been identified as a significant risk factor for depression and suicidality in individuals diagnosed with autism [56,57], whereas strong social ties are known to be protective [38,58–60]. This aligns with previous research showing that connectedness contributed to reduced suicide rates [61]. Our findings suggest that OD may foster such connectedness and thus contribute to a protective context against suicidality.

In line with previous studies [28,29], our findings indicate that individuals diagnosed with autism experienced an increase in trust when deeper issues were collectively explored, fostering a sense of togetherness and mutual understanding. This process, supported by the transparency of therapists and the central role of reflection moments, contributed to a sense of reassurance and recognition. These elements reduced anxiety, enhanced openness to others' perspectives, and supported a shared understanding and collaborative decision-making process in the client-network member-therapist

triad, as aspired in OD [14,62]. Although OD does not prioritize solutions [63], it enables clients and their networks to explore next steps together [64,65], which aligns with the nonlinear nature of recovery [66]. Participants described how feeling understood, receiving real-time feedback, and being actively involved in meaning-making increased their sense of agency, self-confidence, and hope. This resonates with the broader literature on recovery and OD, which emphasizes the importance of meaningful connections, personal empowerment, and a sense of trust and control over one's life as drivers of transformative change [50,66–68]. A noteworthy and novel finding was the significance participants attributed to the seating arrangement, a practice commonly used in OD [69]. Although initially experienced as uncomfortable, the circular format was found to foster a transparent and egalitarian atmosphere.

Furthermore, our results emphasized the sense of being treated as equals by OD therapist, describing a feeling of being seen as human beings rather than as 'clients' or 'problems.' This was associated with reduced stigma, as also noted in prior OD research [29]. Given the common experience of social exclusion in autism, which often contributes to a sense of disconnection [70], this emphasis on mutual respect and equality is especially valuable.

Another aspect that emerged from the results was the facilitating role of OD therapists, which appeared to be particularly valuable to participants given the unique social communication challenges associated with autism. It is known that they have more difficulty with intuitive social attunement and are often aware of this [71]. Participants observed that OD therapists manage the flow of the dialogue, encourage participation, and monitor non-verbal signals, and naturally ensure a fair distribution of speaking turns. It appeared that this may alleviate some of the potential social discomfort.

Interestingly, group size did not emerge as a barrier, contrary to common assumptions about overstimulation in autism. Participants cited various reasons: for some, it was valuable to hear and give meaning to the same story collectively; for others, the OD therapists' guiding role made the process more manageable.

The final point of discussion concerns the concept of "tolerating uncertainty," which is central to OD [18] and evoked mixed reactions. Remarkably, the majority of the participants managed uncertainty well, benefiting from the slow pace: slowing down to avoid quick conclusions or solutions, emphasizing the need to acknowledge and validate everyone's presence early in the session, and making space for their specific stories [18]. It seems that the concept of "tolerating uncertainty" applies to individuals diagnosed with autism just as it does to others, even though difficulty with uncertainty is often observed in individuals diagnosed with autism [72]. The attitude of the OD therapists [18] appeared to contribute to feelings of safety, thus enabling individuals to tolerate uncertainties. Their extensive experience in working with people diagnosed with autism may also have contributed to this. Additionally, reducing miscommunication, a known stressor for individuals diagnosed with autism [73], may have further supported this tolerance. Nevertheless, for some, this feeling of uncertainty seemed to have significantly contributed to their negative sentiments regarding the OD approach.

Although this study shows promising results regarding the suitability of OD for individuals diagnosed with autism, it is important to consider that participants were of average to high cognitive level and possessed good verbal competence. Many individuals diagnosed with autism experience distress even with slight uncertainty, which can exacerbate difficulties with social interaction [74–76]. Tolerance of uncertainty is also known to vary with features of the autism phenotype, with severity of ASD predicting levels of intolerance [77,78]. Interventions that rely heavily on verbal expression in conversation are inherently social processes, requiring building rapport, engaging in personal disclosure, and navigating complex social interactions [79–81]. Executive functioning difficulties, social communication challenges, and high levels of alexithymia further complicate engagement in such therapeutic interventions [82,83]. These considerations highlight that the positive experiences reported in this study may not fully generalize to individuals with lower cognitive or verbal abilities or those with more severe ASD traits.

Reflecting on practical implications, it may be beneficial to consider adaptations to the OD approach to make it more comfortable for individuals who experience difficulties with managing uncertainty. One suggestion that emerged from the data was to better prepare participants by providing a clearer picture of what to expect during the network meeting, as well as encouraging them to reflect in advance on topics they may wish to discuss. Such preparatory steps could enhance

participants' sense of agency and potentially support the recovery process. Similar recommendations regarding the value of preparatory guidance have been made in previous research [30].

### 4.1. Limitations and strengths

As we shed light on the areas where our study faced challenges, recruiting participants proved to be a challenge, resulting in 12 participants out of 36 potential participants. These challenges may be related to the fact that the concerned clients are dealing with severe mental health issues, and OD is offered during a phase of (potential) crisis and/or unrest in the network. This could influence the willingness and/or ability to participate in research. Additionally, time constraints may have played a role. If we had more time after the focus groups, which were limited due to logistical and time constraints associated with study requirements of one of the involved researchers, we might have been able to encourage clients who had canceled their attendance to the focus group to still participate in the study, possibly through individual interviews. This approach could have increased the number of participants slightly. Furthermore, inherent to our study is the potential source of selection bias due to the self-selection of clients. All eligible clients were invited, but of course they were free to decide whether they wanted to participate. Related to self-selection, clients who held strong opinions, whether positive or negative, may have been more inclined to participate. However, our assessment is that maximum diversity has been achieved, as indicated by the expression of both positive and negative opinions. As described in the Study Design section, data saturation was assessed within the research site and was indicated by the observation that no new aspects emerged in the final focus group.

It should be noted that the study's response rate was approximately 33%, meaning that the majority of potential participants did not take part. The perspectives of non-responders may differ from those reported here, and some may have shared features with the minority of participants who expressed negative views of OD. However, the anonymized and aggregated results were presented to the treatment team, who affirmed their consistency with the team's broader experiences, providing no additional feedback. This suggests that the findings reasonably reflect the client group within the studied setting. Still, given the heterogeneous nature of autism [43], generalization to the wider autism population should be made with caution, and further research is warranted.

Nevertheless, in our opinion, our findings hold significant value, as they serve as a starting point for a deeper exploration, shedding light on voices that, heretofore, have remained underexplored in the literature. We consider the collaboration among researchers with diverse backgrounds, including those with and without expertise in autism, as well as those trained in OD and those not, to be a notable strength of this study. This collaborative approach facilitated comprehensive research, resulting in a dataset rich in valuable insights and enabling us to effectively integrate the perspectives of individuals diagnosed with autism into key themes within the field of OD. Additionally, the study's strength lies in its inclusion of both positive and negative attitudes towards the OD approach.

### 4.2. Conclusion and future research

In conclusion, this study suggests that recovery-oriented, person-centered, and network-based OD care can be a viable option for individuals diagnosed with autism, indicating that they are not exceptions despite the specific challenges associated with autism. However, it is essential to recognize the diverse responses to the OD approach. While most participants acknowledged its potential benefits, some expressed strong negative sentiments, indicating that the approach may not resonate with everyone. Therefore, conducting personal assessments remains crucial to ensure that the OD approach aligns with the individual needs of those diagnosed with autism, as is standard for all clients.

Our results suggest that OD care can be offered to individuals diagnosed with autism, although this should be stated with caution due to the small sample size and study design limitations. At the same time, it should be noted that participants in this study were of average to high cognitive level and possessed good verbal competence. Consequently, the experiences reported here may not fully generalize to individuals with lower cognitive or verbal abilities or higher severity

of ASD traits, for whom OD may not be the most appropriate form of care. Future research should further explore this promising avenue, examining the applicability, effectiveness, and nuances of the OD approach within a broader spectrum of the population diagnosed with autism. Additionally, it would be valuable to investigate how preparation for network meetings impacts the experiences of clients engaging with the OD approach, as suggested by some participants in this study.

## Acknowledgments

We thank all participants for participating this study.

## Author contributions

**Conceptualization:** Karin Lorenz-Artz, Marieke Kneepkens.

**Data curation:** Karin Lorenz-Artz, Marieke Kneepkens.

**Formal analysis:** Karin Lorenz-Artz, Marieke Kneepkens.

**Methodology:** Karin Lorenz-Artz, Marieke Kneepkens.

**Project administration:** Karin Lorenz-Artz, Marieke Kneepkens.

**Supervision:** Joyce Bierbooms, Inge Bongers.

**Writing – original draft:** Karin Lorenz-Artz, Marieke Kneepkens.

**Writing – review & editing:** Joyce Bierbooms, Inge Bongers.

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
