## [Decision Letter · Decision Letter 0]

19 Mar 2025

PMEN-D-24-00472

Exploring Open Dialogue and autism: A qualitative client-perspective study

PLOS Mental Health

Dear Dr. Karin Lorenz-Artz

Thank you for submitting your manuscript to PLOS Mental Health. After careful consideration, we believe that your work has significant merit but does not fully meet PLOS Mental Health’s publication criteria in its current form. Therefore, we invite you to submit a revised version of the manuscript that addresses the points raised during the review process.

We would like to extend our apologies for the delay in providing this decision. It has been particularly challenging to secure reviewers, and we appreciate your patience throughout this process. 

Below, we provide comments from the reviewer. These comments, included at the end of this letter, outline the required and recommended changes to strengthen your manuscript. Please review them carefully and provide a point-by-point response in your revised submission.

We look forward to receiving your revised manuscript.

Kind regards,

María Soledad Burrone, PhD, MPH, MD

Academic Editor

PLOS Mental Health

Journal Requirements:

https://journals.plos.org/mentalhealth/s/figures

https://journals.plos.org/mentalhealth/s/figures#loc-file-requirements 

Reviewers' comments:

Reviewer's Responses to Questions

**Comments to the Author**

1. Does this manuscript meet PLOS Mental Health’s publication criteria?

Reviewer #1: Yes

2. Has the statistical analysis been performed appropriately and rigorously?

Reviewer #1: N/A

3. Have the authors made all data underlying the findings in their manuscript fully available (please refer to the Data Availability Statement at the start of the manuscript PDF file)?

Reviewer #1: No

4. Is the manuscript presented in an intelligible fashion and written in standard English?

Reviewer #1: Yes

Reviewer #1: Thank you for the opportunity to review this manuscript. Support services for adults on the autism spectrum is an important area of research that needs more attention. Overall, I found this manuscript a little hard to understand. It would be helpful if the authors could seek advice from an editor to minimise repetition and ensure that the English words used accurately reflect the intended meaning.

Introduction

It would be helpful to move the explanation of what Open Dialogue is on Page 4 to the second paragraph in the Introduction. The audience needs to know what Open Dialogue is before reading about its efficacy.

Reference No. 22 discussed that there is a lack of high quality evidence about Open Dialogue. This should be mentioned in the Introduction.

P8 Line 112: Are the "promising results from Finland" published or part of a manuscript in preparation? It would be good to include a citation.

Method

What does usual care in the MHC involve for clients diagnosed with autism?

Why is the Open Dialogue meeting only offered when the treatment team observes or anticipates a crisis? Can a client choose to take part in the Open Dialogue meetings if they do not have a crisis?

The section on recruitment process can be shortened and made clearer.

Although Open Dialogue meetings do not have a pre-defined agenda, it would be good to have some information on what was usually discussed in the Open Dialogue meetings that the participants attended.

Were Open Dialogue meetings, interviews, and focus groups conducted in person or online?

There is no need to include the names of the themes in the Method section if they are already described in Results.

It would be good to add more information on how the research team's background and existing knowledge influenced the analysis process.

It's a bit unclear how many researchers were involved in each stage of the analysis process, as the plural "researchers" is used throughout but P15 mentions a second researcher.

Results

It would be helpful to reduce the number of quotes and use them selectively to illustrate ideas.

Discussion

Some parts of the Discussion feel a bit repetitive. It would be helpful to re-organise this section to ensure each idea is discussed once.

It would be helpful to suggest how the Open Dialogue approach might be adapted to make it more comfortable for people who have trouble managing uncertainty.

**Do you want your identity to be public for this peer review?** For information about this choice, including consent withdrawal, please see our Privacy Policy

Reviewer #1: No

---

## [Decision Letter · Decision Letter 1]

18 Aug 2025

PMEN-D-24-00472R1

Exploring Open Dialogue and autism: A qualitative client-perspective study

PLOS Mental Health

Dear Dr. Lorenz-Artz,

Thank you for submitting your revised manuscript to PLOS Mental Health.

As explained in a prior email, we needed to send your paper to one more reviewer. They have now returned their report (below and attached) and I would like to offer you one final, minor revision. I will then assess the revised manuscript in-house to save time and given the minor nature of the revisions needed. If you have any questions at all, feel free to reach out to me. 

We look forward to receiving your revised manuscript.

Kind regards,

Dr Karli Montague-Cardoso

Executive Editor

PLOS Mental Health

Journal Requirements:

Additional Editor Comments (if provided):

Reviewers' comments:

Reviewer's Responses to Questions

**Comments to the Author**

Reviewer #1: All comments have been addressed

Reviewer #2: (No Response)

publication criteria?

Reviewer #1: Yes

Reviewer #2: Yes

3. Has the statistical analysis been performed appropriately and rigorously?

Reviewer #1: N/A

Reviewer #2: N/A

4. Have the authors made all data underlying the findings in their manuscript fully available (please refer to the Data Availability Statement at the start of the manuscript PDF file)?

Reviewer #1: No

Reviewer #2: No

5. Is the manuscript presented in an intelligible fashion and written in standard English?

Reviewer #1: Yes

Reviewer #2: Yes

Reviewer #1: (No Response)

Reviewer #2: This paper by Lorenz-Artz and colleagues provides a qualitative exploration of the Open Dialogue method is an autism care setting. The authors utilise focus group discussions and individual interviews to explore client perspectives of this model of care. It is unique in that it gives voice to autistic individuals who historically are often not directly heard from in research.

ABSTRACT

Satisfactory.

INTRODUCTION

Satisfactory as revised.

METHODS

It might be helpful to further clarify how clients for OD are selected. Were there explicit criteria or it is left to individual judgement? There is some content in this regard in lines 152-155 – perhaps the authors might want to bring that up to align with lines 146-148.

Line 167: for ‘gather’ read ‘gathering’

Study Design: the reviewer wonders if the investigators would like to explicitly situate their study within standard qualitative research method. 1) Triangulation – this is the practice of utilising more than one approach (as done in this paper) in order to arrive at more robust findings. It might have been desirable, albeit noting the constraints of willing participants, to have different individuals participate in the FGD and individual interviews. 2) Saturation – this is the practice of gathering qualitative data until additional units do not confer incremental information. Although mention is made of this under the discussion, the means by which this is said to have been achieved is rather unconvincing. 3) How were the focus groups moderated?

Line 201: delete ‘an’

There appears to be an inconsistency between the study design and the details provided under recruitment (lines 214-222). Perhaps the authors might want to include and align the additional detail under study design.

To what extent might the responders on interview have been influenced by prior participation in focus groups?

RESULTS

Satisfactory as revised.

DISCUSSION

Given the cognitive profile of study participants, who were of average to high cognitive level and with good verbal competence, it might be worth discussing how well this model of care is suited to other autistic individuals.

This also applies to the theme of uncertainty in relation to the autism phenotype. Is there a possibility that the challenges with uncertainty might be related to key features of the autism phenotype?

It would have been good to have more discussion of non-responders, and whether their reasons for not participating may represent shared features with the seeming minority of responders who had negative views of OD. Given a response rate of approximately 33 per cent, the attitudes of the majority who constitute non-responders might have a lot to say about the larger autism community.

**Do you want your identity to be public for this peer review?** For information about this choice, including consent withdrawal, please see our Privacy Policy

Reviewer #1: No

Reviewer #2: No

---

## [Editor Report · Decision Letter 2]

16 Sep 2025

Exploring Open Dialogue and autism: A qualitative client-perspective study

PMEN-D-24-00472R2

Dear Dr. Lorenz-Artz,

We are pleased to inform you that your manuscript 'Exploring Open Dialogue and autism: A qualitative client-perspective study' has been provisionally accepted for publication in PLOS Mental Health.

Best regards,

Karli Montague-Cardoso

Staff Editor

PLOS Mental Health